# Resistance pattern of infected chronic wound isolates and factors associated with bacterial resistance to third generation cephalosporins at Mbarara Regional Referral Hospital, Uganda

**Wangoye Khalim** [1,2]*, **James Mwesigye**[3], **Martin Tungotyo**[4,5], **Silvano Samba Twinomujuni**[1]

**1** Department of Pharmacy and Pharmacology, Mbarara University of Science and Technology, Mbarara City, Uganda, **2** Department of Pharmacy, Kiboga general Hospital, Kiboga Town Council, Kiboga, Uganda, **3** Department of Medical Laboratory Science, Mbarara University of Science and Technology, Mbarara City, Uganda, **4** Department of Surgery, Mbarara Regional Referral Hospital, Mbarara City, Uganda, **5** Department of Surgery, Mbarara University of Science and Technology, Mbarara City, Uganda

* khalimwags@gmail.com

## Abstract

### Background

The objectives of this study were; (I) to determine the proportion of pathogens isolated from patients with infected chronic wounds in the surgical ward of MRRH that are resistant to the third-generation cephalosporins and (II) to determine the factors associated with resistance to third-generation cephalosporins in the surgical ward of MRRH.

### Method(s)

This study was a descriptive analytical survey of bacterial isolates from infected chronic wounds among patients admitted in the surgical ward of MRRH, Uganda. Seventy five (75) study participants were recruited in the study using convenient sampling technique. Bacterial culture and identification was performed using standard microbiology laboratory procedures whereas broth microdilution method was used to establish the susceptibility of the identified pathogens. Data for objective one (1) was summarized as proportions while the categorized variables were analyzed using logistic regression to determine whether they were associated with the resistance patterns. The level of significance was preset at 5% and *p*-values less than 0.05 were considered statistically significant.

### Results

Generally, all isolates had complete susceptibility (100%) to Cefoperazone+Sulbactam 2g except 7.1% of *proteus spp* that were resistant. Of all the bacterial isolates studied, *Staphylococcus aureus*, *Enterobacter agglomerans*, *providencia spp* and *pseudomonas earuginosa* had complete resistance (100%) to Cefopodoxime 200mg while *providencia spp* and

**Data Availability Statement:** All relevant data are within the paper and its Supporting Information files.

**Funding:** The authors received no specific funding for this study.

**Competing interests:** The authors have declared that no competing interests exist.

**Abbreviations:** ANOVA, Analysis of Variance; CLSI, Clinical Laboratory Standard Institute; HIV, Human Immunodeficiency Virus; MIC, Minimum Inhibitory Concentration; MUST, Mbarara University of Science and Technology; MRRH, Mbarara Regional Referral Hospital; PCR, Polymerase Chain Reaction; *SPP*, Species; SD, Standard Deviation; REC, Research Ethics Committee; CI, Confidence Interval; OR, Odds ratio.

*pseudomomas earuginosa* had complete resistance (100%) to Cefixime 400mg and cefotaxime 1g. Finally, higher odds of bacterial resistance to more 2 brands of the third generation cephalosporins were observed among participants who had prior exposure to the third generation cephalosporins (OR, 2.22, 95% CI, 0.80–6.14), comorbidities (OR, 1.76, 95% CI, 0.62–4.96) and those who had more than two hospitalizations in a year (OR, 1.39, 95% CI 0.46–4.25). However, multivariate logistic regression was not performed since no factor was significantly associated with resistance to more than two brands of third generation cephalosporins (p >0.05).

## Conclusion

This study found that cefixime and cefpodoixme had high rates of resistance and should not be used in routine management of infected chronic wounds. In addition, the factors investigated in this study were not significantly associated with bacterial resistance to more than two brands of third generation cephalosporins.

## Background

Globally, the burden of infected chronic wounds is likely to increase due to the rising levels of bacterial resistance to antibiotics and non-communicable diseases such as Diabetes mellitus and cancer [1]. In the United States of America alone, more than 6.5 million chronic wounds with evidence of bacterial infection are diagnosed every year [2].

Several studies in Uganda and the rest of East Africa indicates that overused antibiotics such as ceftriaxone have become less effective in treating severe bacterial infections and there is a need for establishing the local knowledge of antibiotic resistance pattern to guide the selection of appropriate antibiotic therapy [3].

In addition, infection of chronic wounds with antibiotic resistant bacterial pathogens slows wound healing and the use of an effective topical or parenteral antibiotic therapy has been recommended as one of the treatment strategies [4].

However, routine culture and sensitivity tests and periodic antibiotic resistance surveillance studies are rarely performed in Mbarara Regional Referral Hospital due to inadequate microbiology supplies and the turnaround time for culture and sensitivity tests is high in the majority of Hospitals in Uganda, causing delays in making clinical decisions required for selection of effective antibiotic therapy [5].

Moreover, evidence-based antibiotic guidelines for management of infected chronic wounds are currently unavailable in the surgical Ward of Mbarara Regional Referral Hospital, making the selection of effective antibiotic therapy impractical.

Consequently, patients with infected chronic wounds are likely to experience long hospital stays, high treatment costs, further delay of wound healing, development of severe invasive bacterial infections and increased emergency of antibiotics resistance if ineffective antibiotics are used.

Therefore this study was conducted to generate third-generation cephalosporins susceptibility map and to understand the factors driving emergency of bacterial resistance so as to guide Clinicians to make evidence-based empirical prescription of third generation cephalosporins required for timely and effective management of infected chronic wounds on the Surgical Ward of MRRH as well as strengthening antibiotic stewardship practices in MRRH.

## Methods

### Study design

The study was a descriptive analytical survey of bacterial isolates from infected chronic wounds at the surgical ward of MRRH from August 2020 to October 2020.

**Study setting.** Participants were enrolled from the surgical ward of MRRH between August 2020 and October 2020. MRRH is a public health facility with a bed Capacity of 300 beds and it is a Regional Referral Hospital in Western Uganda located in Mbarara Municipality, approximately 250 km from Kampala, the capital City of Uganda. Its catchment population is approximately 10 million people. This Hospital also serves as a Teaching Hospital for MUST. According to the patient discharge register, the average number of patients admitted with a diagnosis of various infected wounds is 35 patients per month with an annual prevalence of 420 patients. The surgical ward is currently run by 10 nurses, 25 residents, 4 general surgeons, 1 plastic surgeon, 1 orthopedic surgeon, 1 pharmacy technician,1 Hospital Pharmacist and 1 neurosurgeon. The surgical ward is subdivided into the male and female sections with a total bed capacity of 55.

The microbiological procedures were carried out in the Microbiology Laboratory of MUST, Mbarara City, Uganda. The Microbiology Laboratory is managed by three highly experienced staff that is 2 Laboratory technologists and 1 senior laboratory technologist. The Laboratory is well equipped with the necessary equipment and materials including equipment such as electronic microscopes, florescent microscopes, biosafety cabins, incubators, autoclaves, analytical profile index analyzer and polymerase chain reaction (PCR) machine. Therefore, the Laboratory is able to offer a range of laboratory tests such as gram staining, microscopy, culture and sensitivity tests, Liver function tests and serological tests such as typhoid test, Brucella agglutination test and Human immunodeficiency test (HIV) serology.

In addition, this Microbiology Laboratory follows stringent Laboratory quality assurance measures from the Central public health Laboratory of Uganda that have been designed based on the recommendations of Clinical Laboratory standard Institute.

### Study population

The study population was patients with infected chronic wounds admitted at the Surgical Ward of MRRH in Uganda.

### Study variables

**Dependent variable.** Sensitivity pattern of bacterial isolates to the third-generation cephalosporins.

**Independent variables.** The type of pathogen, prior use of a third-generation cephalosporin, length of Hospital stay, frequency of Hospitalization, comorbidity and patient demographics (age, gender, level of education, occupation and employment).

### Selection criteria

All inpatients admitted in the surgical ward with signs and symptoms of infected chronic wounds (increasing pain at the ulcer site, erythema, edema, heat, purulent exudate, serous exudate, delayed ulcer healing, discolored granulation tissue, friable granulation tissue, wound base pocketing, foul odor and wound breakdown) and consented to participate in the study were included in this study while patients without record of medication history and those who expressed voluntary withdrawal were excluded.

## Sample size of patients with infected chronic wound

The following assumptions were made during sample size calculation;

a. Research data was collected for 3 months and the expected population of patients with infected wounds was 105 patients in 3 months (approximately 35 patients per month). This was based on a review of primary data from the patient discharge register in the surgical ward which had 420 patients with a diagnosis of various infected wounds in one year (2018).

b. The prevalence of infected chronic wounds was estimated to be 22% [6].

$N_o = Z^{2*}P (1-P)/E^2$

$N_o$ = Sample size.

$Z$ = Confidence level.

$P$ = Estimated proportion of infected chronic wounds in the population.

$E$ = Desired level of precision.

$Z$ = 1.96.

$P$ = 0.22(22%).

$E$ = 0.05

$N_o = 1.96^{2*}0.22(1–0.22)/0.05^2$

$= 3.842^*0.22^*0.78/0.0025.$

$= 264$ Patients.

Finite population correction [7]: This was required because the expected average population of patients in three months of data collection was 105 patients based on the above record in the surgical ward.

$n = N_o/ (N_o-1)/N+1,$

n = Adjusted Sample size.

N = Population size (105 patients).

$n = 264/ (264–1)/105+1.$

$n = 264/3.5.$

$= $**75 patients**.

## Sampling technique

Convenience sampling technique was used to select the study subjects who met the criteria for infected chronic wounds [8].

## Variable measurement and study procedures

For diagnosis of infected chronic wounds and assessment of known patient associated factors of bacterial resistance to the third generation cephalosporins, a checklist containing symptoms and signs of chronic infected wounds was used by the Clinician to guide the clinical diagnosis of chronic infected wounds as well as assessment of associated factors of bacterial resistance to the third generation cephalosporins.

## Sample collection and bacterial identification

Two nurses working in the surgical ward were trained by an experienced laboratory technologist to empower them with skills of obtaining wound swabs for culture and sensitivity.

After obtaining an informed consent from the patients meeting the criteria, routine clinical samples were aseptically collected by a trained nurse from the patients' wound base using sterile cotton swabs. The standard operating procedure developed by British Columbia Provincial

Nursing Skin & Wound Committee were used to ensure an aseptic procedure [9]. The samples were transported to the Microbiology Laboratory of MUST within 30 minutes. Only one swab was obtained from each patient after cleaning the wound base with sterile normal saline.

## Laboratory procedures

I. Primary cultures: On receipt, swab specimens were registered in the laboratory research register.

II. Depending on the nature of samples, each specimen was inoculated on chocolate, blood, mannitol salt sugar, xylose lysine decarboxylated agar, and MacConkey Agar as follows; and incubated at

III. Using inoculating loop, each sample was streaked onto the upper one fourth portion of an agar plate with parallel overlapping strokes. The plates were labeled.

IV. The loop was flamed and allowed to cool. The plate was turned at right angle. Overlapped the previous streak once or twice and repeated the streaking process on one-half of the remaining area.

V. Procedure 4 was repeated.

VI. The plates were incubated overnight at 35˚C-37˚C in the incubator.

VII. After incubation for 16–20 hours, the plates were checked for bacterial growth.

VIII. Representative bacterial colonies were selected based on the difference in shape, size and color. Selected colonies from each plate were sub-cultured and incubated overnight.

IX. Bacterial identification: This was performed based on morphological, cultural characteristics such as hemolysis on blood agar, swarming (positive for *proteus spp*), changes in physical appearance on differential agar (pink appearance of lactose-fermenting bacterial colonies on macConkey agar), motility test was positive for *enterobacter agglomerans* and *providencia spp*. In addition, Table 1 shows the biochemical tests that were performed to confirm the identity of bacterial pathogens;

**Table 1. Biochemical tests for identification of bacterial pathogens.**

| Isolate | Biochemical test | Expected results |
|---|---|---|
| *Staphylococcus aureus* | Catalase | Positive |
| | Coagulase | Positive |
| | Mannitol fermentation | Positive |
| | Dnase | Positive |
| *Klebsiella spp* | Citrate | Positive |
| | Urea | Positive |
| | Indole | Negative |
| *Proteus spp* | Hydrogen sulphide | Positive |
| | Urea | Positive |
| | Citrate | Positive |
| | Oxidation | Positive |
| *Enterobacter agglomerans* | Hydrogen sulphide | Negative |
| | Urea | Negative |
| | Indole | Negative |
| *Providential spp* | Indole, methyl red, citrate, nitrate reductase and catalase | Positive |

## Antibacterial susceptibility testing

The minimum inhibitory concentrations and antibacterial susceptibility testing were performed using broth microdilution technique as described by CLSI and the review in the general principle and practices of antimicrobial susceptibility testing [10]. The Procedure for Broth microdilution involved the following steps;

X. Preparation of stock solutions: Stock solutions were prepared based on the manufacturer's instruction for reconstitution. All the 5 antibiotic brands did not have potency information and the weight for antibiotics were calculated based on the highest plasma concentrations derived from the following pharmacokinetic studies because of the correlation that exist between MIC and pharmacokinetic parameters [11]. Table 2 shows the weight of antibiotics as calculated based on their respective maximum plasma concentrations.

I. Using a pipette, 100μl of sterile brain heart infusion were dispensed into the wells of microtitre plates, each row labeled to corresponding antibiotic.

II. 100μl of the antibiotic stock solution were also dispensed into the well in column 1. Using the pipette set at 100μl, mix the antibiotics into the wells in column 1 by sucking up and down 6 times.

III. 100μl of this were withdrawn from column1 and added to column 2, making column 2 a two-fold dilution of column 1.

IV. 100μl of column 2 were transferred to column 3. This was repeated down to column 9.

V. 5μl of isolates suspended in sterile water and adjusted to McFarland turbidity ($10^4$x$10^5$CFU/ml) were dispensed into the wells except wells in column 11 for sterility control. Wells in column 10 were used for growth control and contained 100μl of brain heart infusion and 5μl of isolates.

VI. Microtitre plates were then covered with sterile aluminum foil to prevent evaporation during incubation.

VII. After 24 hour incubation at 37˚C, the microtitre plates were observed using a reading mirror for visible bacterial growth as indicated by turbidity and a measure of bacterial resistance to the third generation cephalosporins.

**Table 2. Weight of powder for stock solutions.**

| S/no. | Antibiotic. | Maximum plasma concentration (desired concentration). | Reference. | Weight of powder(g) (desired concentration) X volume of diluent (1000ml) divide by 1000000 |
|---|---|---|---|---|
| 1 | Ceftriaxone 1g (Epicephin®) | 168μg/ml | [12]. | 0.168g |
| 2 | Cefoperazone+ Sulbactam 2g (Sulcef®) | 159μg/ml | [13] | 0.159g |
| 3 | Cefotaxime 1g (Omnatax®) | 41.1μg/ml | [14]. | 0.0411g |
| 4 | Cefpodoxime 200mg (Ximeprox®) | 2.7μg/ml | [15] | 0.0027g |
| 5 | Cefixime 400mg (gramocef-o 400®). | 2.47μg/ml | [16] | 0.00247g |

The antibiotic solutions were kept in the refrigerator at a temperature of 4˚C.

## Quality control

To ensure consistent and high quality research outputs, the researcher implemented quality control measures throughout the entire research process. Antibiotics for the third generation cephalosporins, culture media and staining reagents were procured from premises licensed by the National Drug Authority of Uganda to avoid the risk of counterfeit products which could affect the quality of research results. In- addition, the procured antibiotics, culture media and staining reagents were strictly stored at conditions specified by the manufacturers to avoid product deterioration during the research process.

## Data processing and analysis plan

The study data was entered into Microsoft Excel and exported to STATA version 15.0 for statistical analysis. Frequencies, and mean (SD; standard deviation) were computed to summarize the data.

- **Objective 1:** The susceptibility data of bacterial isolates was summarized as proportions and presented in a group bar chart.

- **Objective 2:** In addition, the categorized variables were analyzed using logistic regression to determine whether they were associated with the resistance patterns. The final results were presented in a table.

    The level of significance was preset at 5% and *p*-values less than 0.05 were considered statistically significant in each of the above statistical tests.

## Ethical approval

This study was approved by the research ethics committee of Mbarara University of science and technology (**Protocol registration number: 06/12-19**). In addition, all methods were performed in accordance with the relevant guidelines/regulations and informed consent was obtained from all participants or legal guardians.

## Results

### Demographic and clinical characteristics of respondents

Table 3 presents the general characteristics of 75 study participants who were diagnosed with infected chronic wounds on the surgical ward of MRRH. The table shows that 43 (57.3%) participants were below 40 years old and the mean age for all the participants was 40.7 years (SD = 16.4). The mean length of hospital stay was 8.23 days (SD = 4.67) and the mean frequency of hospitalization was approximately twice in a year. Furthermore, 43 (57.3%) of the study participants had no prior exposure to third generation cephalosporins.

**Resistance patterns of bacterial isolates from infected chronic wounds.** Fig 1 shows susceptibility profile of Six bacterial species isolated from chronic infected wounds of patients (n = 69/75). Overall, the studied bacterial isolates from chronic wounds were most resistant to Cefopodoxime 200mg (Ximeprox®) and Cefixime 400mg (gramocef-0-400®) with overall resistance rates ranging from 90–100% and 70–100% respectively. Generally, all isolates had complete susceptibility (100%) to Cefoperazone+Sulbactam 2g except 7.1% of *proteus spp* that were resistant. Of all the bacterial isolates studied, *Staphylococcus aureus*, *Enterobacter agglomerans*, *providencia spp* and *pseudomonas earuginosa* had complete resistance (100%) to Cefopodoxime 200mg (Ximeprox®) while *providencia spp* and *pseudomomas earuginosa* had complete resistance (100%) to Cefixime 400mg and cefotaxime 1g. *Proteus species and*

**Table 3. Demographic and clinical characteristics of respondents.**

| Characteristics | Level | Overall (n = 75) |
|---|---|---|
| Age group (years) | <40 | 43 (57.3) |
| | ≥40 | 32 (42.7) |
| | mean (SD) | 40.7(16.4) |
| Sex | Female | 33 (44) |
| | Male | 42 (56) |
| Educational level | Post-primary | 20 (27) |
| | Primary | 38 (50) |
| | post-secondary | 17 (23) |
| Type of employment | Formal employment | 15 (20) |
| | Informal employment | 31 (41.3) |
| | None employed | 29 (38.7) |
| Length of Hospital stay | Mean(SD) | 8.23 (4.67) |
| Frequency of hospitalization per year | Mean (SD) | 1.9 (0.8) |
| Comorbidity | No | 49 (65.3) |
| | Yes | 26 (34.7) |
| Prior exposure to third generation cephalosporins | No | 43 (57.3) |
| | Yes | 32 (42.7) |

*providencia spp* were most resistant to Ceftriaxone 1g (66.7% and 100% respectively). The least resistant bacterial isolate to most brands (2/5) of third generation cephalosporins investigated in this study was *Enterobacter agglomerans*.

**Factors associated with bacterial resistance to the third generation cephalosporins.**
Table 4 below shows results of bivariate logistic regression analysis of factors associated with resistance to third generation cephalosporins among patients with infected chronic wounds in the surgical ward of MRRH. Resistance to more than two third generation cephalosprin brands was considered as the primary outcome.

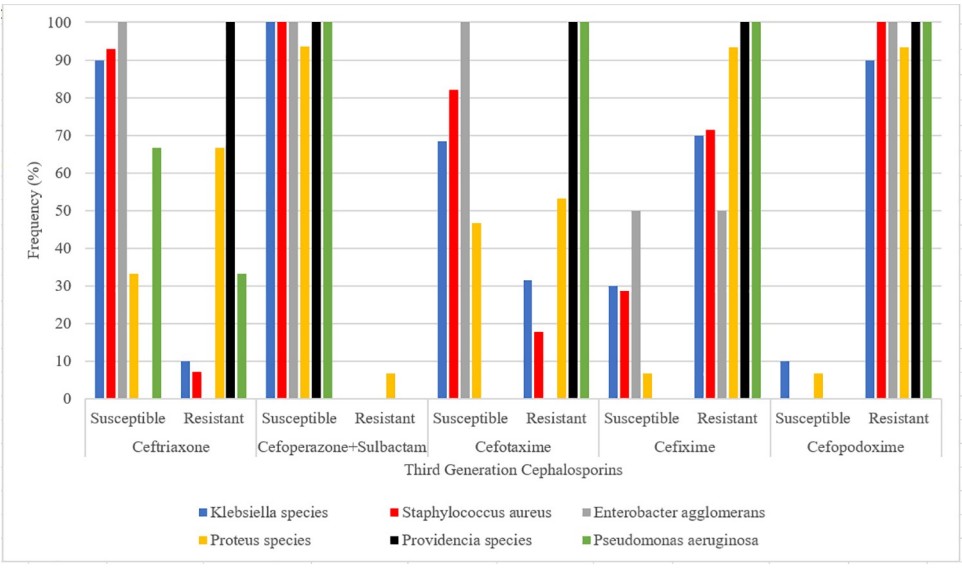

**Fig 1. Overall susceptibility profile of infected chronic wound isolates against selected third generation cephalosporins.**

**Table 4. Bivariate logistic analysis of differences in resistance to third generation cephalosporins.**

| Variable | n (%) | Resistance, n (%) | | Crude Odds Ratio (95% CI) | P value |
|---|---|---|---|---|---|
| | | ≤ 2 drugs | > 2 drugs | | |
| Age (years) | | | | | |
| ≤ 40 | 49 (56.5) | 28 (71.8) | 11 (28.2) | | |
| > 40 | 30 (43.5) | 18 (60.0) | 12 (40.0) | 1.70 (0.62–4.66) | 0.305 |
| Sex | | | | | |
| Female | 31 (44.9) | 18(58.1) | 13 (41.9) | | |
| Male | 38 (55.1) | 28 (73.7) | 10 (26.3) | 0.49 (0.18–1.36) | 0.174 |
| Length of hospital stay | | | | | |
| ≤ 7 days | 36 (52.2) | 23 (63.9) | 13 (36.1) | | |
| > 7 days | 33 (47.8) | 23 (69.7) | 10 (30.3) | 0.77 (0.28–2.11) | 0.610 |
| Frequency of Hospitalization in a year | | | | | |
| ≤ 2 times | 51 (73.9) | 35 (68.6) | 16 (31.4) | | |
| > 2 times | 18 (26.1) | 11 (61.1) | 7 (38.9) | 1.39 (0.46–4.25) | 0.562 |
| Prior-exposure to 3GC | | | | | |
| No | 39 (56.5) | 29 (74.4) | 10 (25.6) | | |
| Yes | 30 (43.5) | 17 (56.7) | 13 (43.3) | 2.22 (0.80–6.14) | 0.125 |
| Comorbidity | | | | | |
| No | 45 (65.2) | 32 (71.1) | 13 (28.9) | | |
| Yes | 24 (34.8) | 14 (58.3) | 10 (41.7) | 1.76 (0.62–4.96) | 0.286 |

Higher odds of bacterial resistance to more 2 brands of the third generation cephalosporins were observed among participants who had prior exposure to the third generation cephalosporins (OR, 2.22, 95% CI, 0.80–6.14), comorbidities (OR, 1.76, 95% CI, 0.62–4.96) and those who had more than two hospitalizations in a year (OR, 1.39, 95% CI 0.46–4.25). However, multivariate logistic regression was not performed since no factor was significantly associated with resistance to more than two brands of third generation cephalosporins (p >0.05).

## Discussion

With respect to bacterial resistance against the third generation cephalosporins, infected chronic wound isolates exhibited the highest rates of resistance ranging from 70% to 100% (Fig 1) against cefixime (gramocef-0-400[R]) and cefpodoxime (Ximeprox[R]). Findings from previous similar studies revealed comparable resistance of infected chronic wound isolates ranging from 87.6% to 100% resistance against cefpodoxime (Ximeprox[R]) and cefixime (gramocef-0-400[R]) [17, 18]. Therefore the therapeutic benefit of cefixime (gramocef-0-400[R]) and cefopoxime 200mg (Ximeprox[R]) is extremely low to manage infected chronic wounds and continued use of these antibiotics will burden the patients with long hospital stays, high treatment costs, further delay of wound healing and development of severe invasive bacterial infections [19].

All isolates had complete susceptibility (100%) against cefoperazone+sulbactam 2g (Sulcef[R]) except *proteus spp* which exhibited 7.1% resistance. Similar susceptibility studies from other clinical settings also reported no resistance of infected chronic wound isolates (*staphylococcus aures* and *Klebsiella spp*) against cefoperazone+sulbactam [20, 21], therefore cefoperazone+sulbactam can be recommended as the empirical therapy for management of severe infected chronic wound isolates because of its low overall rate of resistance.

It is important to note that Cefoperazone+Sulbactam was the most effective third generation cephalosporins and this could be attributed to sulbactam, a beta-lactamase inhibitor is

capable of inhibiting growth for most pathogens producing beta-lactamase enzyme that inactivates beta-lactam drugs such as cephalosporins [21].

It was also observed that *Proteus spp* and *Providencia spp* exhibited the highest rates of resistance against ceftriaxone 1g (66.7% and100% respectively, Fig 1). This finding is in agreement with results from other clinical settings that presents even a much higher prevalence of *proteus mirabilis* resistance against ceftriaxone of (83.8%) [22] In light of the above study, ceftriaxone can still be used on the surgical ward of MRRH for the treatment of chronic wound infected with *proteus spp* after confirmation of culture and sensitivity results.

Based on crude odds ratio resulting from bivariate logistic regression analysis (Table 4), patients who had prior exposure to third generation cephalosporins, comorbidities, age less than 40 years and multiple hospitalizations in a year are more likely to develop resistance to more than two brands of third generation cephalosporins. However, the associations ware not statistically significant (P>0.05) for all the factors analyzed. Comparatively, studies elsewhere have demonstrated a strong statistically significant relationship between antibiotic resistance and length of hospital stay, prior antibiotic exposure and multiple hospitalization [23–25].

## Conclusion

This study found that cefixime gramocef-0-400®) and cefpodoixme 200mg (ximeprox®) had high rates of resistance and should not be used in routine management of infected chronic wounds. Infected chronic wound isolates had least resistance to Cefoperazone+salbactam 2g (Sulcef®) and can be used as empirical therapy in management of infected chronic wounds. In addition, the factors investigated in this study were not significantly associated with bacterial resistance to more than two brands of third generation cephalosporins.

## Supporting information

**S1 Dataset.**
(XLSX)

## Author Contributions

**Conceptualization:** Wangoye Khalim.

**Formal analysis:** Wangoye Khalim.

**Investigation:** James Mwesigye.

**Methodology:** James Mwesigye.

**Project administration:** Wangoye Khalim.

**Resources:** Wangoye Khalim.

**Supervision:** Martin Tungotyo, Silvano Samba Twinomujuni.

**Validation:** Silvano Samba Twinomujuni.

**Writing – original draft:** Wangoye Khalim.

**Writing – review & editing:** Martin Tungotyo, Silvano Samba Twinomujuni.

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
