## [Decision Letter · Decision Letter 0]

26 Oct 2021

PONE-D-21-22072

Resistance pattern of infected chronic wound isolates and factors associated with bacterial resistance to third generation cephalosporins at Mbarara Regional Referral Hospital, Uganda.

PLOS ONE

Dear Dr. Khalim,

Thank you for submitting your manuscript to PLOS ONE. After careful consideration, we feel that it has merit but does not fully meet PLOS ONE’s publication criteria as it currently stands. Therefore, we invite you to submit a revised version of the manuscript that addresses the points raised during the review process.

We look forward to receiving your revised manuscript.

Kind regards,

Grzegorz Woźniakowski, Full professor, PhD, ScD

Academic Editor

PLOS ONE

“This study had no funding support from any organization/Institution. The corresponding author used out-of- pocket to fund this study as part of  his postgraduate studies at Mbarara University of Science and Technology.

All the remaining authors did not receive any grant to support this study but contributed non-financial support to this study.”

The following authors had no financial and non-financial competing interest;

1. W.K

2. J.M

3. T.M

4. T.S.S

Additional Editor Comments (if provided):

Reviewers' comments:

Reviewer's Responses to Questions

**Comments to the Author**

1. Is the manuscript technically sound, and do the data support the conclusions?

Reviewer #1: Yes

2. Has the statistical analysis been performed appropriately and rigorously? 

Reviewer #1: Yes

3. Have the authors made all data underlying the findings in their manuscript fully available?

Reviewer #1: Yes

4. Is the manuscript presented in an intelligible fashion and written in standard English?

Reviewer #1: Yes

5. Review Comments to the Author

Reviewer #1: Overall this was a good study that focused on a matter of great public health importance.

While the study objectives are listed in the abstract there is no mention of the objectives in the methodology sections.

It is not clear whether the authors differentiated between hospital or community acquired chronic wound infections given that this may influence antibiotic resistance patterns. While there was a consideration for ≤2 prior hospitalizations it is not clear whether “0” or no hospitalization was also evaluated.

While the background section mentions some comorbidities as factors contributing to bacterial resistance the authors do not state what comorbidities the study participants were evaluated for.

Despite there being no single factor with significant The statistical analysis and inferences made are however valid.

There are a few editorial issues that could be easily resolved by the authors e.g some minor typos with regard to the use of upper or lower case letters with regard to nouns. Another grammatical error that could be corrected is in paragraph 2 of the section titled “Factors associated with bacterial resistance to the third generation cephalosporins”. The first sentence of in this second parapraph reads “Higher odds of bacterial resistance to more 2 brands of the third generation cephalosporins….”. Did the authors mean to sate “2 brands or more”, or “more than 2 brands”? This should be clarified.

6. PLOS authors have the option to publish the peer review history of their article (what does this mean?). If published, this will include your full peer review and any attached files.

Reviewer #1: No

---

## [Author Response · Author response to Decision Letter 0]

10 Nov 2021

Response to Reviewers:

Preamble: this document outlines the changes/revisions made as a result of review process made. All revisions have been written in red colour. This document consists of two main sections (journal requirements and Review Comments to the Author). The authors have also made the following minor changes;

-The phrase ‘’infected chronic wound’’ is changed to “chronic wound infection” throughout the revised manuscript.

- Additional affiliation (5) was added in the revised manuscript were the first author works as a head of department.

(a)Journal Requirements:

1. Please ensure that your manuscript meets PLOS ONE's style requirements, including those for file naming. The PLOS ONE style templates can be found at https://journals.plos.org/plosone/s/file?id=wjVg/PLOSOne_formatting_sample_main_body.pdf and https://journals.plos.org/plosone/s/file?id=ba62/PLOSOne_formatting_sample_title_authors_affiliations.pdf.

Response: All major section headings are level headings; bold with font size 18 throughout the revised manuscript, sub-headings are level 2 headings, bold and font size 16 while headings under subheadings are level 3 headings, bold and font size 14. All first letters of the headings are now capitalized. We also introduced cells in table 4 as per the above guidelines. See page 16. Equations have been restated using equation tool (page 6-7). Figure 1 has been removed from the manuscript body and submitted separately. Figure title and legend have been stated on page 14.

Title, authors and affiliations have been revised as per PLOS ONE guidelines above. 

“This study had no funding support from any organization/Institution. The corresponding author used out-of- pocket to fund this study as part of his postgraduate studies at Mbarara University of Science and Technology. 

All the remaining authors did not receive any grant to support this study but contributed non-financial support to this study.”

Response: The authors have revised the above statement and the new statement of financial disclosure is “The authors received no specific funding for this work.” See page 20 under declaration section.

3.Please include your amended statements within your cover letter; we will change the online submission form on your behalf.

 Response: Amendments have been stated in the cover letter and Witten in red color in the cover letter.

The following authors had no financial and non-financial competing interest;

1. W.K

2. J.M

3. T.M

4. T.S.S

Response: we have revised the above statement and the new statement is “The authors have declared that no competing interests exist” see competing interest on page 20 of the revised manuscript with track changes highlighted in red color.

Response: The corresponding author has authenticated his pre-existing ID in the editorial manager. The ORCID ID is 0000-0002-5446-9352.

Response: The ethics statement has been removed ‘’declaration sections’’ and ‘’selection criteria’’ and now appears only in methods section of the revised manuscript with track changes on page 4 under methods section.

Response: Reference list has been reviewed. It is complete and correct.

(b) Review Comments to the Author

1. While the study objectives are listed in the abstract there is no mention of the objectives in the methodology sections. It is not clear whether the authors differentiated between hospital or community acquired chronic wound infections given that this may influence antibiotic resistance patterns.

Response: We (the authors) have clarified on page 5 under study population we studied community acquired chronic wound infection. This has been stated also in objective 1 on page 12 under subsection “data processing and analysis plan”

2. While there was a consideration for ≤2 prior hospitalizations it is not clear whether “0” or no hospitalization was also evaluated.

Response: we only considered patients with at least 1 hospitalization in a year. See a caption under table 4 has been included to clarify this on page 16.

3. While the background section mentions some comorbidities as factors contributing to bacterial resistance the authors do not state what comorbidities the study participants were evaluated for.

Response: We studied patients with chronic wound infection who had diabetes mellitus and hypertension. This has now been included under subsection “selection criteria’’ on page 6 of the revised manuscript.

4. There are a few editorial issues that could be easily resolved by the authors e.g some minor typos with regard to the use of upper or lower case letters with regard to nouns.

Response: We have made various corrections regarding minor typos with regard to the use of upper or lower case letters with regard to nouns, particularly names of drugs, isolates and other nouns throughout the revised manuscript. See highlighted the first letters of various nouns that have been capitalized e.g. page 1 under subsection ’’results’’ of the abstract and page 14 under subheading “resistance pattern of chronic wound isolates”

5. Another grammatical error that could be corrected is in paragraph 2 of the section titled “Factors associated with bacterial resistance to the third generation cephalosporins”. The first sentence of in this second parapraph reads “Higher odds of bacterial resistance to more 2 brands of the third generation cephalosporins….”. Did the authors mean to sate “2 brands or more”, or “more than 2 brands”? This should be clarified.

Response: The grammatical error has been resolved. We have stated “more than 2 brands” see paragraph 2 on page 15 under subsection “factors associated with bacterial resistance to third generation cephalosporins”

---

## [Editor Report · Decision Letter 1]

26 Nov 2021

Resistance pattern of infected chronic wound isolates and factors associated with bacterial resistance to third generation cephalosporins at Mbarara Regional Referral Hospital, Uganda.

PONE-D-21-22072R1

Dear Dr. Khalim,

We’re pleased to inform you that your manuscript has been judged scientifically suitable for publication and will be formally accepted for publication once it meets all outstanding technical requirements.

Kind regards,

Grzegorz Woźniakowski, Full professor, PhD, ScD

Academic Editor

PLOS ONE
---

## [Editor Report · Acceptance letter]

6 Dec 2021

PONE-D-21-22072R1 

Resistance pattern of infected chronic wound isolates and factors associated with bacterial resistance to third generation cephalosporins at Mbarara Regional Referral Hospital, Uganda. 

Dear Dr. Khalim:

I'm pleased to inform you that your manuscript has been deemed suitable for publication in PLOS ONE. Congratulations! Your manuscript is now with our production department. 

Kind regards, 

on behalf of

Prof. Grzegorz Woźniakowski 

Academic Editor

PLOS ONE